# Preparation and Dispersion Performance of Hydrophobic Fumed Silica Aqueous Dispersion

**DOI:** 10.3390/polym15173502

**Published:** 2023-08-22

**Authors:** Jinglu Xu, Jihu Wang, Shaoguo Wen, Shengnan Ding, Jia Song, Sihong Jiang, Haopeng Wang

**Affiliations:** College of Chemistry and Chemical Engineering, Shanghai University of Engineering Science, Shanghai 201620, China; xujinglu2015@163.com (J.X.); wangjihu@sues.edu.cn (J.W.); 15266859266@163.com (S.D.); palpitate0627@163.com (J.S.); tony8906@126.com (S.J.); 13293087591@163.com (H.W.)

**Keywords:** fumed silica, dispersant, dispersibility, particle size, waterborne resin

## Abstract

Hydrophobic fumed silica (HFS) is a commonly used rheology additive in waterborne coatings. A series of experiments were conducted on the HFS-dispersing technology in this study. The size and structure of HFS primary particles were observed via transmission electron microscopy (TEM). The measurement results of the TEM were D_50_ = 13.6 nm and D_90_ = 19.7 nm, respectively. The particle size and dispersion performance of HFS were tested via dynamic light scattering (DLS). Additionally, the HFS aqueous dispersion was prepared and compounded with waterborne polyacrylic latex and polyurethane resin. The elemental distribution of the coatings was characterized using energy dispersive spectroscopy (EDS). The results show that the HFS in a non-ionic polymer dispersant had the best dispersion performance. The particle size of the HFS in the aqueous dispersion is related to the dispersion conditions. Under optimized conditions, the HFS aqueous dispersion can be prepared with a particle size of D_50_ = 27.2 nm. The HFS aqueous dispersion has stable storage stability. Even after storage for 47 d, the particle size still did not change significantly.

## 1. Introduction

The colloidal stability greatly affects the application performance of nanoparticles in coatings, adhesives, and rubber, such as anti-corrosion, and mechanical and optical properties [1,2,3]. Under the background of rapidly rising energy prices, how to shorten the colloidal stability time and reduce the energy consumption of nanoparticle dispersion in production has become a problem that relevant enterprises must consider.

Fumed silica is a kind of amorphous particle with a small size, approximately between 10 and 40 nm, and an extremely large specific surface area [4]. In addition to providing thermal insulation, thickening, thixotropy, and anti-settling [5,6,7,8], it also has the effect of reinforcement, color development, and improving powder fluidity [9,10,11,12,13]. Fumed silica has been widely used in coatings, adhesives, and other composite materials [14,15,16,17,18,19]. Hydrophobic fumed silica (HFS) is an aerosol with alkanes, chloromethyl groups, or other hydrophobic functional groups on the surface [20]. Under the joint action of hydrophobic functional groups and hydrogen bonds, a three-dimensional network structure can be formed in a highly polar system, which could provide better rheological (thixotropy) and mechanical properties for waterborne coatings and adhesives compared to other fumed silica [4,21,22]. However, the hydrophobic functional groups also made HFS extremely difficult to disperse in water [23].

Lei and Kang [24,25] explored surface modification methods that made it possible to obtain slurry with nanoparticle size distribution at lower energy inputs using HFS. Actually, changes in processes indirectly increased the costs and limited application prospects. Schilde [26] conducted systematic research on the impact of traditional colloidal stability on the particle size of silica slurry. It is obvious that the particle size of the silica slurry obtained via grinding is the smallest, which is due to the high energy conversion efficiency of grinding. During the grinding process, the frequent collision of beads effectively breaks the physical bond and chemical bond between the nanoparticles [27]. The grinding process still has the drawbacks of consuming high energy and generating significant heat. If the production unit is not equipped with a cooling device, it may lead to catastrophic consequences in large-scale production. Although high-speed dissolvers have low energy conversion efficiency [26,27], their relatively low energy input and cheap equipment still make them one of the mainstream dispersion methods. In recent years, some research has shown that redesigning the blades of a dissolver can significantly improve its dispersion performance [28,29]. The energy consumption of ultrasonic dispersion is between grinding and high-speed dispersion [27]. However, the dispersion efficiency of ultrasonic dispersion is much higher than that of high-speed dispersion, which is close to grinding [26]. The combination of high-speed dispersion and ultrasonic dispersion may be able to reach a particle size close to milling while reducing energy consumption.

Due to its high solid content, HFS slurry is convenient for transportation and storage, while HFS aqueous dispersion is more convenient for direct use. In this work, a two-step method was used to prepare HFS slurry and HFS aqueous dispersion. We found that using high-speed dispersion combined with ultrasonic dispersion will help disperse HFS into nanoscale aqueous dispersion. Firstly, an appropriate dispersant was added to the HFS. Under a high shear rate, the slurry was prepared from the mixture. Then, the slurry was added into the aqueous dispersion system to dilute it into a certain proportion, and ultrasonic dispersion was used in the preparation of the HFS aqueous dispersion. To explore the optimized process, the influencing factors of the colloidal stability were systematically studied, including dispersant type, dispersant addition amount, and dilution concentration. Additionally, the variation trend of HFS particle size with ultrasonic dispersion time was analyzed. Finally, the HFS aqueous dispersion was compounded with waterborne resin, and EDS mapping was used to evaluate the distribution of HFS in different coatings.

## 2. Materials and Methods

### 2.1. Materials

The hydrophobic fumed silica was acquired from Wacker AG and the basic properties are shown in Table 1. N,N-Dimethylethanolamine (DMEA, 98%) was purchased from Aladdin Reagent (Shanghai) Co., Ltd. Chemically pure sodium hexametaphosphate and sodium lauryl benzenesulfonate were purchased from Sinopharm Chemical Reagent Co., Ltd. Anionic polymer dispersant and cationic polymer dispersant were provided by Anhui Guangcheng New Material Technology Co., Ltd. Non-ionic polymer dispersant was purchased from Yihua High-tech (Nanjing) Co., Ltd. Waterborne polyacrylic latex and waterborne polyurethane resin were purchased from Shanghai Qixiangqingchen Coating Technology Co., Ltd. and Covestro Polymer Co., Ltd., respectively. All the above reagents were used as received.

### 2.2. Preparation of HFS Slurry and Aqueous Dispersion

The preparation process of HFS slurry and aqueous dispersion is shown in Figure 1.

The slurry was prepared using a high-speed dissolver (Shanghai Pushen Testing Instrument Co., Ltd., Shanghai, China; blade diameter, 7.0 cm). A certain amount of deionized water and dispersant were added into a vessel with low-speed stirring at 400 rpm. DMEA was used to adjust the solution’s pH to 8.5. The mixture was stirred for 5 min. At a low speed of 300 rpm, 20 g of HFS was added in divided doses. After all the HFS was added, the shear rate was set at 4000 rpm, and the mixture was stirred at a high speed for 5 min to prepare the HFS slurry with a solid content of 20 wt%. During the dispersion process, the slurry was kept at a constant temperature (25 °C) using water-cooling equipment. The HFS slurries were stored for 2 h for defoaming. After diluting with appropriate deionized water, the slurry was dispersed using an ultrasonic disperser (Grows Instrument Co., Ltd., Shanghai, China; instantaneous power, 200 W; frequency, 40 KHz) to obtain HFS aqueous dispersion. The HFS slurry formulation is shown in Table 2.

### 2.3. Preparation of Waterborne Composite Film

The HFS aqueous dispersion was compounded with different waterborne resins. The coatings were prepared with the following steps: added an appropriate amount of waterborne resin in a container and slowly added HFS aqueous dispersion. The rotation speed was set to 800 rpm and the coatings were stirred for 10 min. After being left for 24 h to defoam, the coating was poured into a PTFE mold (depth: ~500 μm) and dried at 50 °C for 3 days to obtain the dried films with a thickness of around 400~500 μm. The formulations of HFS composite coatings are shown in Table 3.

### 2.4. Characterization

The microscopic morphology and particle size of hydrophobic fumed silica were characterized and analyzed using a microscope (PH-PG3230, Phenix Optics Co., Ltd., Shanghai, China) and a transmission electron microscope (TEM, JEOL-2100F, JEOL, Akishima, Japan). The cross-section of the coatings was swept with an SEM-EDS (Zeiss Sigma 300, Zeiss, Oberkochen, Germany) energy spectrometer to obtain the element distribution.

The contact angle of the dispersant solution on the HFS surface was carried out with a water contact angle tester (Shanghai Pushen Testing Instrument Co., Ltd., Shanghai, China), and this method was taken from reference [30]. Before testing the particle size, the HFS aqueous dispersion was diluted 1000 times with the DMEA solution (pH = 8.5). After a certain amount of time of ultrasonic dispersion, the aqueous dispersion containing HFS was placed in a container to stabilize for 60 s. The particle size and polydispersity index (PDI) of the nanoparticles in the dispersion were tested using the Zetasizer Nano-S90 laser particle size analyzer of Malvern Company. The particle size test was performed on the HFS aqueous dispersion just after completing the ultrasonic dispersion. The testing temperature was set at 25 °C. The average of the three measurements was taken to obtain the final result.

## 3. Results and Discussion

### 3.1. Dispersibility of HFS in Slurry and Aqueous Dispersion

#### 3.1.1. Particle Morphology of HFS

In order to obtain clearer particle size results, the microscopic morphology and particle size distribution of HFS were observed via TEM. The results are shown in Figure 2.

The TEM image in Figure 2 can more clearly characterize the shape and particle size of HFS. It can be seen from Figure 2a,b that HFS particles have many pore structures. HFS agglomerates have particle sizes around 200 to 300 nm [31]. It is formed by several 50~150 nm aggregates connected by “bridges” [32,33]. These so-called “bridges” are actually formed by physical bonds between the edges of the aggregate and other aggregates (Figure 2d). The particle size distribution curves of HFS obtained via data fitting are shown in Figure 2c. Before dispersion, the HFS was formed by many aggregates with a size of about 200~300 nm. The aggregates were linked by physical bonds, such as hydrogen bonds, London forces, and other interaction forces, forming a larger agglomerate. These physical bonds could be reduced by surface property change and broken by high shear force. It can be seen from Figure 2c that the distribution curve of the particle size number of HFS primary particles is normally distributed, with D_50_ = 13.6 nm. However, the strong chemical bonds could hardly be broken by dispersing; it would take more energy to break the aggregates into primary particles. Ideally, HFS particles may only be dispersed to a particle size of 50~150 nm without any chemical bond breakage [32].

#### 3.1.2. The HFS Dispersibility with Different Dispersants

In order to study the impact of different dispersants on HFS dispersibility, several representative dispersants were selected. According to the formula in Table 2, HFS was dispersed by using dispersant A (sodium hexametaphosphate), dispersant B (sodium dodecyl benzene sulfonate), dispersant C (cationic polymer dispersant), dispersant D (sodium polycarboxylate, anionic polymer dispersant), and dispersant E (non-ionic polymer dispersant). When the addition of the dispersant was lower than 6 wt%, no matter which dispersant was used, the HFS slurry seemed like gel. The results are shown in Figure 3.

From Figure 3, it can be confirmed that the HFS slurries prepared with dispersants A, B, and C are white or yellowish bulk solids, and HFS cannot be wetted and dispersed in the aqueous phase system. The HFS slurry prepared by using dispersants D and E is a light-yellow liquid. The slurry prepared by using these two dispersants has good fluidity, indicating that HFS particles are sufficiently wetted. For the former, HFS has a strong adsorption effect on the long-chain alkyl of the dispersant [34]. The negatively charged carboxyl group of the dispersant is exposed to the solution and forms a solvation chain. The non-ionic polymer dispersant was adsorbed on the surface of the particles through hydrophobic chain segments [35]. However, hydrophilic chain segments extend in the solution, forming a steric effect to prevent agglomeration between particles [29,36,37].

In order to accurately distinguish the wetting state of different dispersant solutions on the surface of HFS particles, the water contact angle was measured. The results are shown in Figure 4.

Due to the alkylation of the HFS surface, water is totally unable to wet the HFS surface, and the water contact angle is even larger than 170°. After the addition of the dispersant (A, B, or C), the wetting condition was significantly improved, and the contact angle was reduced to 110~130°. With the addition of dispersant D, the contact angle was further reduced to 56°, indicating a transition from non-wettable to wettable. The water contact angle of non-ionic dispersant E is only 46°, and the wetting performance is significantly better than that of dispersant D. Furthermore, a large number of anionic surfactants are easy to aggregate in the waterborne system and form ion channels inside the coatings, which will affect the initial water resistance and long-term durability of the coatings, especially for waterborne industrial coatings with high initial water resistance requirements [38]. Therefore, it is reasonable to use a non-ionic dispersant (E) to disperse HFS.

#### 3.1.3. The Impact of Dispersant Dosage on HFS Dispersibility

In order to study how dispersant E addition affects the particle size of HFS aqueous dispersion, the dilution ratio was set at 10 times. Five minutes of ultrasonic dispersion was performed to obtain sufficient dispersion energy. The particle size is shown in Figure 5.

As can be seen from Figure 5, after adding 2% of dispersant E, the Z-average particle size decreased to 230 nm. In contrast, the HFS was completely unable to disperse without the use of a dispersant. As can be seen from Figure 4, the HFS surface is totally unable to be wetted by water. When the dispersant E addition is around 4~10%, the particle size of the HFS slightly decreases with the increase in usage, showing a nearly linear trend. When the addition of dispersant E reaches 10 wt%, the particle size is the smallest, and the Z-average particle size is 126.4 nm. Although the dispersant addition increased to 12 wt% and 14 wt%, the particle size did not change any further. The reasons are as follows. (1) When the addition of the dispersant increased, the adsorption of the dispersant molecules on the HFS particles surface also increased. At the same time, the physical bond strength between the aggregates was weakened. As a result, the HFS became more and more easily dispersed, and the particle size became smaller after ultrasonic dispersion. (2) The dispersant reached saturation adsorption on the HFS surface [39,40]. The excess dispersant formed micelles in the solution and had no effect on the particle dispersion. Therefore, the particle size no longer decreased. (3) The strong chemical bonds between the primary particles could not be broken by adding a dispersant. However, the dispersant successfully broke the physical bonds. This study also revealed that the chemical bonds could not be broken via high-speed dispersion and ultrasonic dispersion. As shown in Figure 2, the HFS primary particles’ average size is 19.7 nm. After dispersion, the size of the agglomerates only decreased to 126.4 nm. In conclusion, when the amount of dispersant is controlled at 10 wt%, the slurry has the best dispersion effect in a waterborne system. No obvious difference in particle size between each HFS aqueous dispersion (10~14 wt% dispersant addition) means that it is hard to reduce particle size by adding any more dispersant.

It can also be reconfirmed from Figure 6 that the dried HFS powder is a large agglomerate of 5~100 μm. When the amount of dispersant addition is not enough (2 wt%), the solution is not enough to moisten large HFS particles. Only when the dispersant addition exceeds 10 wt% can HFS be fully wetted and dispersed, and finally form a uniform slurry.

#### 3.1.4. The Storage Stability of HFS Aqueous Dispersion with Different Dilution Ratios

The HFS slurry containing 10 wt% dispersant E was diluted 10 times, 100 times, 1000 times, and 10,000 times with the DMEA solution (pH = 8.5). In order to further investigate the storage stability of HFS aqueous dispersion after dilution, the dispersive solution was ultrasonic for 5 min and stayed at a constant temperature (25 °C) and humidity (30%) for 0 d, 3 d, and 47 d. They were used to observe whether there were stratifications, settlements, or other phenomena.

As can be seen from Figure 7, the color of the slurry changed from light yellow to a white turbid zed liquid after 10 dilutions. With the increase in dilution ratio, the dispersion gradually became clear and transparent. After standing for 3 d, the slurry and its diluted dispersion did not change significantly. Even standing for 47 d, stratification, settlement, and other phenomena did not appear in the slurry. This indicates that after the HFS slurry is diluted and dispersed in the waterborne system, the dispersant can be evenly adsorbed on the surface of the particles. Due to the steric hindrance effect between particles increasing, it is difficult to form agglomerates again [41]. Therefore, the storage of the slurries becomes stable. Moreover, it seems that the dilution only has a slight influence on the dispersity and settlement of HFS. Therefore, in order to not add too much water into the coatings, the dilution ratio was selected as 10 times in the preparation of HFS aqueous dispersion.

#### 3.1.5. The Impact of Ultrasonic Dispersion Time on the HFS Dispersibility

To confirm the impact of the ultrasonic dispersing time on the particle size of HFS aqueous dispersion, a study was conducted on this. The dispersant usage and dilution ratio were set at 10 wt% and 10×, respectively. The results are shown in Figure 8.

As can be seen in Figure 8, after high-speed dispersion, the particle size of the HFS slurry was around 1500 nm. After 2 min of dispersion, the particle size of the HFS aqueous dispersion sharply decreased to ~135 nm, which is consistent with the results in the literature [23]. After 10 min of dispersion, the particle size of HFS aqueous dispersion was only 126.4 nm. The dispersion of HFS is a dynamic equilibrium process. From the perspective of dynamics and thermodynamics, this process is mainly divided into three stages: To reduce the surface energy, the initial particle form aggregates with other particles to stay at a low energy state. With the increase in ultrasonic dispersion time (0~2 min), the agglomerates absorb mechanical energy and heat and split into smaller secondary particles. At this stage, the energy provided via ultrasonic dispersion is much larger than that consumed by the dispersion of HFS. When the ultrasonic time is 2~5 min, the curve of the particle size showed a gentle downward trend. This is because the surface area of particles increases rapidly after they split into primary particles, and the increased surface energy ∆G is much larger. When the ultrasonic time is more than 5 min, the particle size of silica has almost no obvious change. This is due to the limited energy provided via ultrasonic dispersion of a certain power. In this state, the agglomeration of aggregate particles forms a dynamic equilibrium with the dispersion of agglomerate particles. Therefore, due to the limited power of ultrasonic dispersion, the particle size gradually decreases with the increase in ultrasonic time but eventually establishes equilibrium and stays steady. From the perspective of reducing energy consumption, it seems that ultrasonic dispersion for 5 min is the most reasonable, as both energy consumption and particle size are already small enough at this time. It can be inferred that, after the ultrasonic dispersion, the particle size will form agglomeration again due to the energy loss of the system. In addition, if the temperature of the system decreases, the settlement speed will become faster. Larger particles will wrap the smaller particles in the settlement process, which will also lead to an increase in particle size. Storage at the slurry status and applying dilution and ultrasonic dispersion before using water dispersion may effectively avoid this problem, which is strongly recommended in this study.

The energy consumed per kilogram of HFS dispersion can be roughly calculated based on the HFS aqueous dispersion mass, ultrasonic dispersion time, and instrument power. Excitingly, the energy consumption of HFS aqueous dispersion prepared in this paper is 120 kJ/kg, which is much lower than the results (165.6 kJ/kg) in reference [26].

#### 3.1.6. Particle Size Distribution of HFS under Optimized Conditions

The optimized process parameters (dilution 10×, ultrasonic dispersion time 10 min) obtained from the above analysis were used to prepare HFS aqueous dispersion. The particle size of the HFS aqueous dispersion was measured by using a laser particle size analyzer. The results are shown in Figure 9.

According to Figure 9a, under the optimal dispersion conditions, the particle size (D_50_) of hydrophobic HFS particles is 27.2 nm and PDI = 0.326. The Z-average particle size is 126.4 nm. However, there is still a gap between the particle size distribution calculated via TEM in Figure 2. Under optimized conditions, the particle size of the agglomerates decreases significantly, even very close to that of the aggregates. However, it still cannot be dispersed to the size of primary particles due to the high energy needed to break chemical bonds. After a 47 d storage, the D_50_ value of HFS aqueous dispersion is 31.0 nm, which is still close to that before storage. The Z-average particle size is 129.8 nm, which is also close to the value before storage. However, the PDI slightly increased, which means that flocculation was happening after storage for 47 d.

### 3.2. Dispersibility of HFS in Coatings

#### 3.2.1. Surface Morphology of Different Waterborne Coatings with HFS

The HFS aqueous dispersion was combined with different waterborne resins. During the process of film formation, the HFS particles tended to form agglomerates in the film due to the decrease in surface tension. It is difficult to characterize the dispersibility of HFS particles in the film-forming process. Therefore, the cross-section of the film was obtained by using the liquid nitrogen brittle breaking method, and the morphology was observed via SEM. The test results are shown in Figure 10 and Figure 11.

As seen in Figure 10a, a few particle aggregates are distributed on the surface of the coating. In Figure 10b, a large number of HFS particles and their agglomerates appeared on the surface of the HFS/waterborne polyurethane composite coating. Due to the poor compatibility between HFS particles and waterborne polyurethane, the particles could hardly be dispersed in the coating. At last, because of the low surface energy, the HFS particles float to the polyurethane coating surface as it dries.

#### 3.2.2. Cross-Sectional SEM of Different Waterborne Coatings with HFS

As can be seen from Figure 11a,b, most of the HFS particles can be evenly distributed in the film, but there are still a few irregular HFS agglomerates in the cross-section. The agglomerates are composed of numerous small HFS aggregates, and the sizes are between 200 and 300 nm. As can be seen in Figure 11c,d, the HFS particles are clumped together to form micron-scale (>5 μm) particles. Based on the analysis of the structure of the film surface, we can draw a conclusion that there is poor compatibility between HFS and waterborne polyurethane. HFS cannot be evenly dispersed in the film. Large aggregates formed by HFS particles exist in the coating, while smaller and lighter particles float to the surface of the coating.

#### 3.2.3. EDS Mapping of Different Waterborne Coatings with HFS

In order to further evaluate the dispersion of HFS in the film, EDS mapping can be used to estimate the characteristic Si element distribution of HFS in the film section. The test results are shown in Figure 12.

The signal distribution of the Si element in the coating section can be visually seen in Figure 12. The distribution of HFS in the aqueous polyacrylic resin film is relatively uniform. The Si signals were concentrated on the HFS agglomerates in the waterborne polyurethane coatings. There are serval reasons for the forming of agglomerates in polyurethane. One is the desorption of the dispersant on the HFS surface. If the interaction force between the polyurethane resin and the dispersant is much stronger than the adsorption between the HFS and the dispersant, the dispersant will tend to desorb from the HFS surface. Additionally, the other one has higher hydrophilia than polyacrylic resin. Although the HFS used in this work is hydrophobic, the HFS particles are hydrophilic after the dispersant adsorption. With more hydroxyl and carboxyl groups, polyacrylic resin is hydrophilic and has better compatibility with HFS slurry.

Based on the above analysis, to obtain the best distribution of HFS in waterborne coatings, it is critical to consider the compatibility between HFS and coatings to avoid the formation of HFS aggregation in the forming process of the film.

## 4. Discussion

In this paper, HFS slurry and aqueous dispersions were prepared, and the dispersion technology was studied. The results of the water contact angle show that the non-ionic dispersant had the best wetting and dispersing effect on HFS particles. To obtain the best dispersion effect, the addition of non-ionic dispersants should be added at 10 wt%. The study on the ultrasonic dispersion process indicates that ultrasonic dispersion for 5 min is the most reasonable. Moreover, it is exciting to note that the HFS aqueous dispersion is stable after using a process that combines high-speed dispersion and ultrasonic dispersion. After storage for 47 d, the particle size remained at D_50_ = 31.0 nm. This shows that the dispersing process without milling is feasible for dispersing HFS. This new process may save energy costs for related enterprises. Moreover, EDS and SEM showed that the dispersion of HFS aqueous dispersion in acrylic resin was uniform and the compatibility was good. This indicates that HFS dispersion has application value.

## Figures and Tables

**Figure 1 polymers-15-03502-f001:**
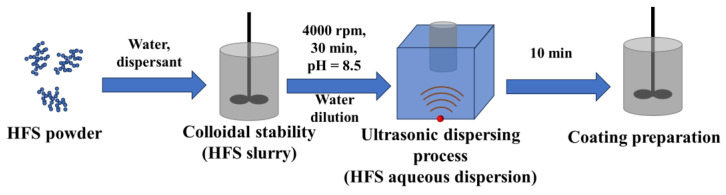
The schematic illustration of HFS aqueous dispersion and coating preparation.

**Figure 2 polymers-15-03502-f002:**
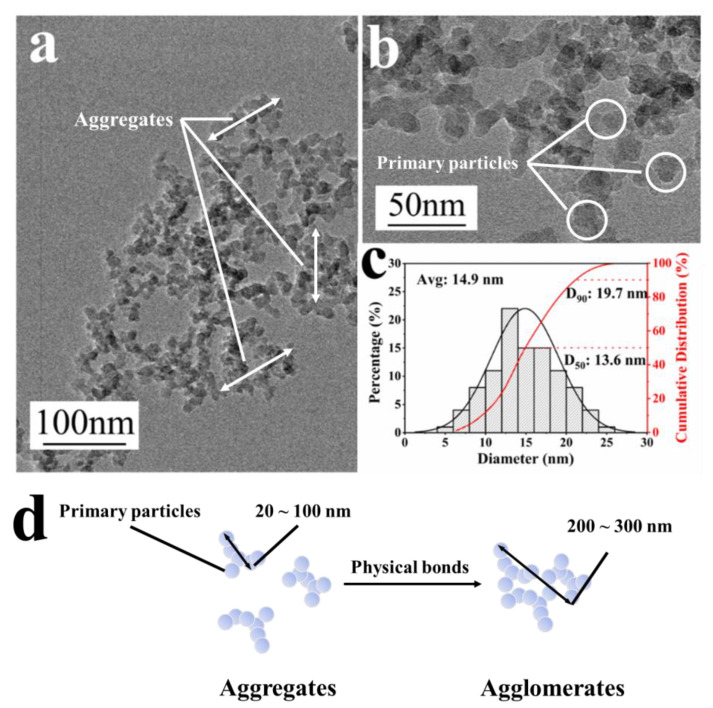
TEM images in different magnifications: (**a**) 30,000×, (**b**) 60,000×, (**c**) particle distribution of HFS dried powder, and (**d**) the schematic illustration of HFS agglomerate formation.

**Figure 3 polymers-15-03502-f003:**
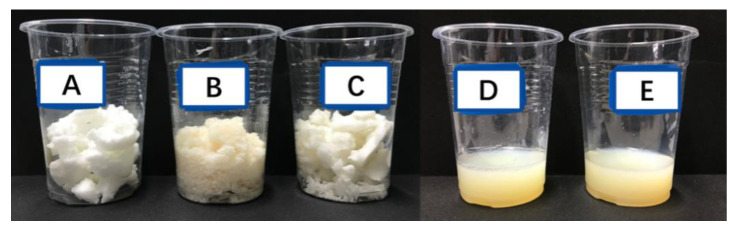
HFS slurry prepared with A (anionic dispersant); B (anionic dispersant); C (cationic polymer dispersant); D (anionic polymer dispersant); and E (non-ionic polymer dispersant).

**Figure 4 polymers-15-03502-f004:**
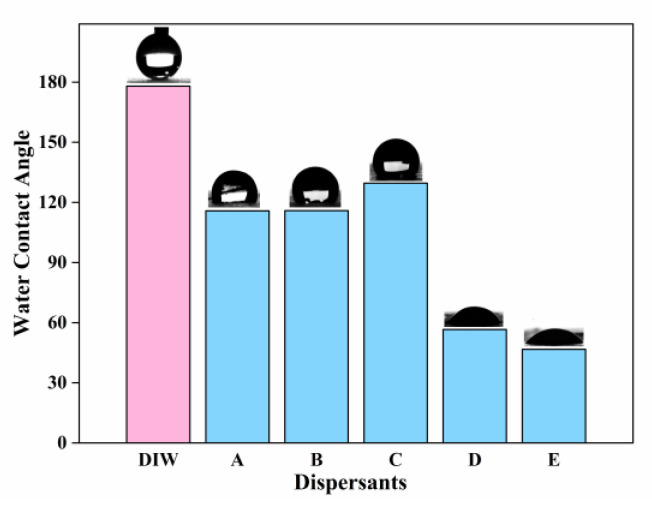
The water contact angle of different dispersant solutions (6 wt%) on the surface coated with HFS powder. A (anionic dispersant); B (anionic dispersant); C (cationic polymer dispersant); D (anionic polymer dispersant); and E (non-ionic polymer dispersant).

**Figure 5 polymers-15-03502-f005:**
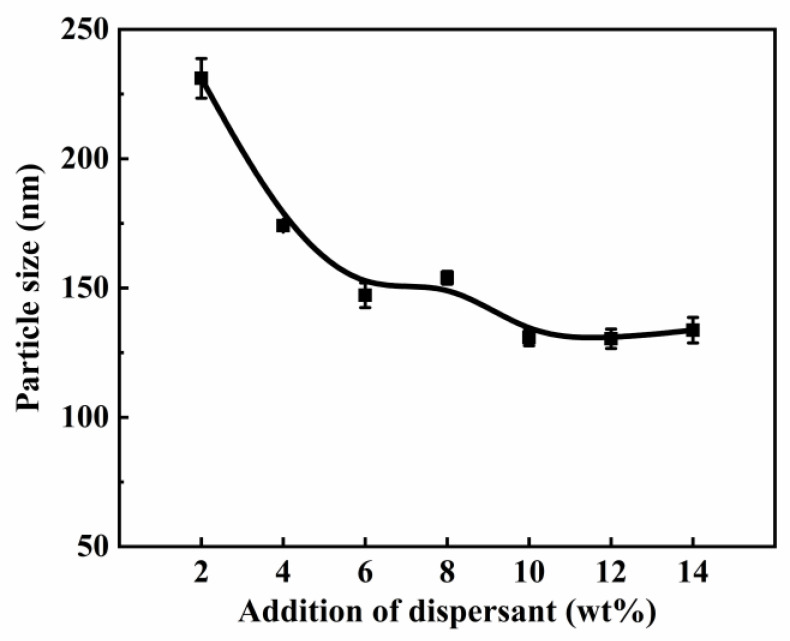
The relationship between the addition of dispersant E and the particle size.

**Figure 6 polymers-15-03502-f006:**
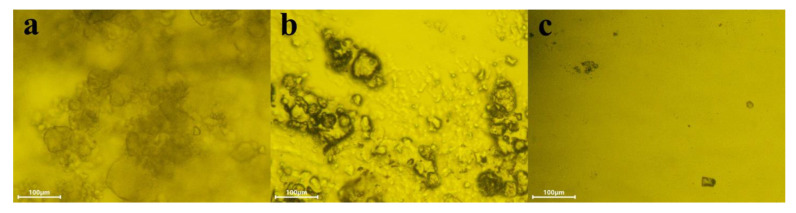
Microscope images of HFS in (**a**) dried powders, (**b**) 2 wt% dispersant solution, and (**c**) 10 wt% dispersant solution.

**Figure 7 polymers-15-03502-f007:**
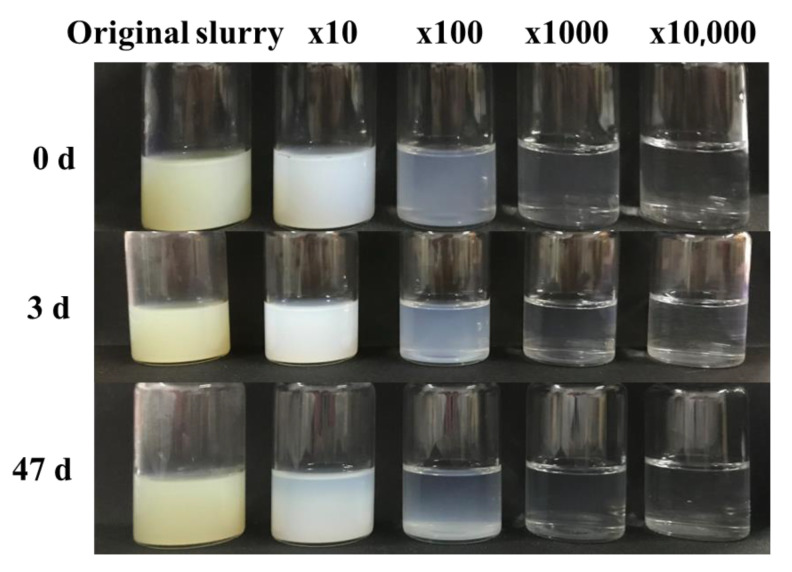
The storage stability of HFS slurry with different dilution ratios.

**Figure 8 polymers-15-03502-f008:**
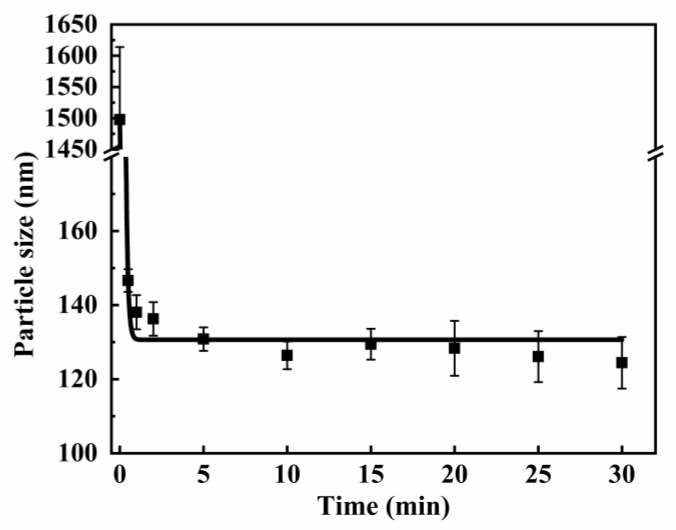
The relationship between the ultrasonic dispersion time and the particle size.

**Figure 9 polymers-15-03502-f009:**
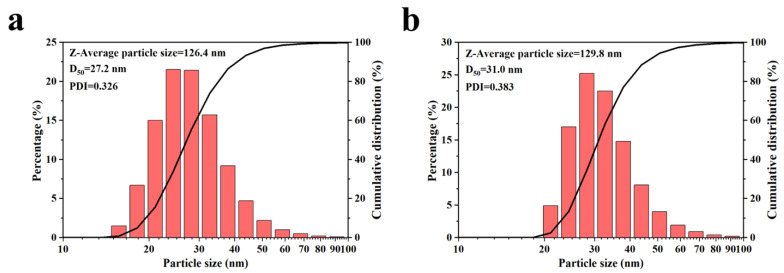
Particle size test results under optimal dispersion conditions: (**a**) storage after 0 d, (**b**) storage after 47 d.

**Figure 10 polymers-15-03502-f010:**
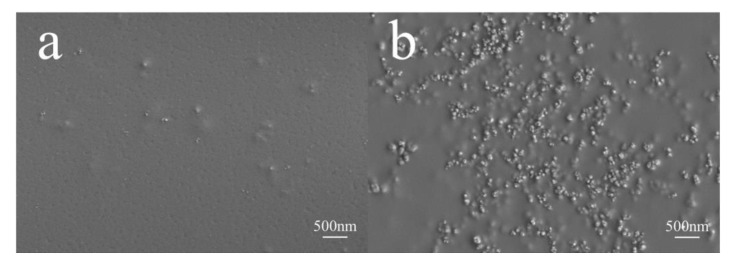
Surface SEM images of HFS with different resin films: (**a**) HFS/waterborne polyacrylic film; (**b**) HFS/waterborne polyurethane film.

**Figure 11 polymers-15-03502-f011:**
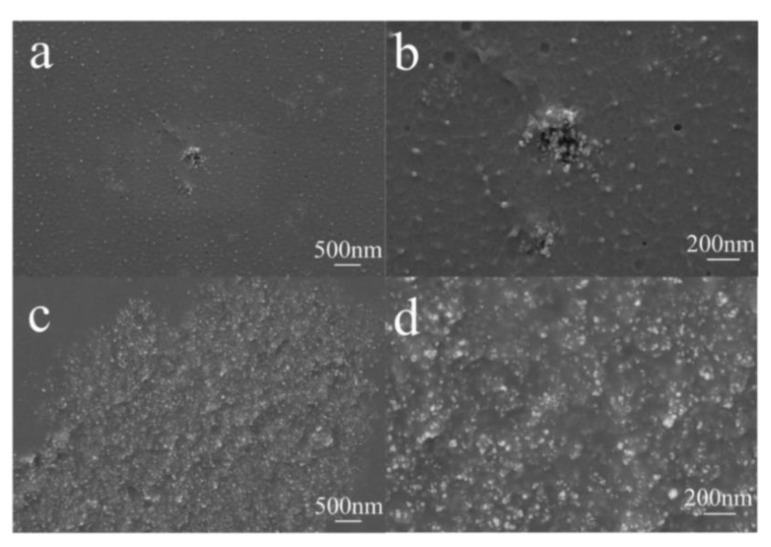
Cross-sectional SEM images of HFS with different resin films: (**a**,**b**) HFS/waterborne acrylic film; (**c**,**d**) HFS/waterborne polyurethane film.

**Figure 12 polymers-15-03502-f012:**
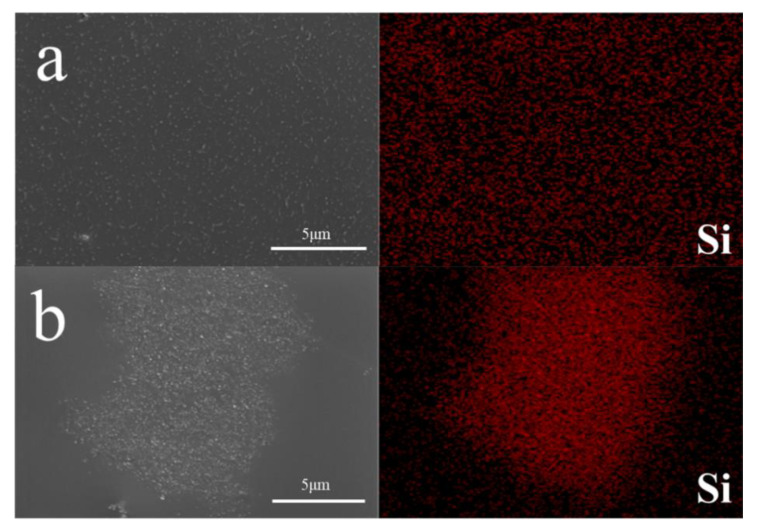
EDS mapping total spectrum and Si spectrum of HFS with different films: (**a**) HFS/waterborne polyacrylic film; (**b**) HFS/waterborne polyurethane film.

**Table 1 polymers-15-03502-t001:** Basic properties of HFS.

Materials	Type	Surface Group	BET-Area/ (m^2^·g^−1^)	Stacking Density/ (g·L^−1^)	Particle Size (nm)
HFS	hydrophobic	Dimethylsiloxy	170 ± 30	40	200~300

**Table 2 polymers-15-03502-t002:** Formula of HFS slurry.

Materials	Mass Fraction/%
Deionized water	78.0; 76.0; 74.0; 72.0; 70.0
Dispersant	2.0; 4.0; 6.0; 8.0; 10.0
HFS	20.0

**Table 3 polymers-15-03502-t003:** Formula of HFS composite coating.

Materials	Mass Fraction/%
Resin (waterborne polyacrylics latex or waterborne polyurethane resin)	98.0
HFS aqueous dispersion (2% HFS)	2.0

## Data Availability

The data presented in this study are available in the article.

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
