# Peer review of "Preparation and Dispersion Performance of Hydrophobic Fumed Silica Aqueous Dispersion"

_polymers, 2023, doi:10.3390/polym15173502_

Round 1

Reviewer 1 Report

In the present work, titled:  Preparation and Dispersion Performance of Hydrophobic Fumed Silica Aqueous Dispersion, the authors analyze how to stabilize and get the right colloidal stability for fumed silica nanoparticles in different media. To reach this objective, they first prepared a slurry of the particles with water and a dispersant, then the slurry was added in two different resins, waterborne polyacrylics latex, and waterborne polyurethane. I believe that the authors need to make some corrections and it might be appropriate to improve the wording and the choice of words to describe the results.

1. The authors used the phrase ´´dispersion process´´ in repeat sentences while in many cases it will be more accurate to replace it with ´´colloidal stability´´, as in the first sentence of the introduction section. The colloidal stability of nanoparticles is the limiting factor in their applications, and the process used to disperse them is the way in which colloidal stability is achieved.

2. Many times, the redaction could be improved by avoiding repeating words or phrases in the same sentence; for example, in lines 31 and 32: Fumed silica is a kind of amorphous particle with a small size and extremely large specific surface area, and its particle size is between 10-40 nm. It could be replaced by: Fumed silica is a kind of amorphous particle with a small size, between 10-40 nm, and extremely large specific surface area.

3. Many times the authors state that the grinding process or the dispersion by stirring under high shear rates effectively breaks physical and chemical bonds. The last is a wrong concept, such it is impossible to break chemical bonds by a physical action such as stirring or grinding, to break this kind of interaction it is needed to add other chemicals to the system, or in the case of a thermo-labile system employ high temperatures. Another fact is that the particles are not bonded by chemical bonds, but instead by a combination of different weak interactions like H-bonds, Van der Waals interaction, dipole-dipole, etc. In a nucleation process, that occurs in bottom-up systems (from chemical synthesis), the primary small particles are redissolved and condensate into bigger particles, and in this case, chemical bonds allow the growth, by it implies a re-dissolution process, not by simply the aggregation of particles (that is a physical phenomenon). In figure 2 is it shown that aggregates are formed by chemical bonds, which is not right.

4. It will be appropriate to include in Table 1 the size of the particle. Then it is the characterization of the particles by microscopy, but in that table where the basic properties are listed, it is important to include the size.

5. Again, regarding redaction, in line 120 authors write: in order to obtain more intuitive particle size results, the microscopic morphology 120 and particle size distribution of HFS were observed by TEM. Intuitive is not the correct word, because when a technique like TEM is used, nothing is intuitive, the image is the reality (under high vacuum conditions), so it could be more accurate to use estimate instead of intuitive. In line 290, the word intuitive is again employed.  In line 139, a point has to be replaced by a comma: ´´ dispersant D (anionic polymer dispersant). and dispersant E (non-ionic polymer dispersant´´.

6. In line 150 authors state that the former dispersant (polycarboxylate) produces anchorage on the surface of HFS particles through strong polar carboxyl groups. It is not clear how it happens due to the negative surface charge of fumed silica particles.

7. Finally, for SEM characterization, figure 9, it will be more accurate to show images with higher zoom. In this case, the scale bar is 200 nm, while non-agglomerates are in that order under high vacuum conditions, as can be observed from the image. 

Authors could check the text and avoid repeating words in the same sentence. 

In the title of section 3.1.2, with is more accurate than at

Author Response

Reviewer 1

In the present work, titled:  Preparation and Dispersion Performance of Hydrophobic Fumed Silica Aqueous Dispersion, the authors analyze how to stabilize and get the right colloidal stability for fumed silica nanoparticles in different media. To reach this objective, they first prepared a slurry of the particles with water and a dispersant, then the slurry was added in two different resins, waterborne polyacrylics latex, and waterborne polyurethane. I believe that the authors need to make some corrections and it might be appropriate to improve the wording and the choice of words to describe the results.

[Response] We thank for the reviewer’s positive comment on our work, and we have made a careful revision according to the reviewer’s suggestion and the details are shown as follows.

  1. The authors used the phrase “dispersion process” in repeat sentences while in many cases it will be more accurate to replace it with “colloidal stability”, as in the first sentence of the introduction section. The colloidal stability of nanoparticles is the limiting factor in their applications, and the process used to disperse them is the way in which colloidal stability is achieved.

[Response] Thanks for the reviewer’s suggestion. The suggestion is helpful. We have revised the “dispersion process” to “colloidal stability” in the below sentences.

“The colloidal stability greatly affects the application performance of nanoparticles in coatings, adhesives, and rubber, such as anti-corrosion, mechanical, and optical properties [1-3]. Under the background of rapidly rising energy prices, how to shorten the colloidal stability time and reduce the energy consumption of nano particle dispersion in production has become a problem that relevant enterprises must consider.”

“Schilde [27] took systematic research on the impact of traditional colloidal stability on the particle size of silica slurry.”

Figure 1. The schematic illustration of HFS aqueous dispersion and coatings preparation

  1. Many times, the redaction could be improved by avoiding repeating words or phrases in the same sentence; for example, in lines 31 and 32: Fumed silica is a kind of amorphous particle with a small size and extremely large specific surface area, and its particle size is between 10-40 nm. It could be replaced by: Fumed silica is a kind of amorphous particle with a small size, between 10-40 nm, and extremely large specific surface area.

[Response] Thanks very much for the reviewer’s suggestion. We have revised the sentences. What’s more, we also found some grammar mistakes in other sentences and revised the mistakes.

  1. Many times the authors state that the grinding process or the dispersion by stirring under high shear rates effectively breaks physical and chemical bonds. The last is a wrong concept, such it is impossible to break chemical bonds by a physical action such as stirring or grinding, to break this kind of interaction it is needed to add other chemicals to the system, or in the case of a thermo-labile system employ high temperatures.

Another fact is that the particles are not bonded by chemical bonds, but instead by a combination of different weak interactions like H-bonds, Van der Waals interaction, dipole-dipole, etc. In a nucleation process, that occurs in bottom-up systems (from chemical synthesis), the primary small particles are redissolved and condensate into bigger particles, and in this case, chemical bonds allow the growth, by it implies a re-dissolution process, not by simply the aggregation of particles (that is a physical phenomenon). In figure 2 is it shown that aggregates are formed by chemical bonds, which is not right.

[Response] Thanks very much for the reviewer’s suggestion.

(1) Regarding to the first question, we fully agree that the high-speed dispersion is physical process and no chemical reaction occurs in this process (unless other reactants are added, or heated at high temperatures until decomposition occurs). This is because the physical bonds energy range is 40 to 50 kJ/mol, and the energy required to break the physical bonds is lower compared to chemical bonds. Energy of the chemical bonds is around 500-1000 kJ /mol, which is much larger than the energy input of high-speed dispersion. However, the energy input and instantaneous power provided by the grinding process is actually larger than the energy of the chemical bonds (ionic bonds or covalent bonds, etc., when the aggregates particles are broken by smashing). In fact, it is easy to understand that when you smash the NaCl crystal with a hammer, the ionic bonds of the NaCl crystal also break. For understanding, we provide the following reference:

“Isao Funahashi, Keita Kondo, Yu Ito, Mina Yamada, Toshiyuki Niwa. Novel contamination-free wet milling technique using ice beads for poorly water-soluble compounds. International Journal of Pharmaceutics, 2019, 563, 413-425”

“Mehdi Kazemimostaghim, Rangam Rajkhowa, Takuya Tsuzuki, Xungai Wang. Production of submicron silk particles by milling. Powder technology, 2013, 241, 230-235.”

“Dereka, B.; Yu, Q.; Lewis, N. H.; Carpenter, W. B.; Bowman, J. M.; Tokmakoff, A. Crossover from hydrogen to chemical bonding. Science, 2021, 371(6525), 160-164.”

(2) Regarding to the second question, the picture we have drawn may be misleading. Primary particles (small crystallites) are tiny crystals formed at the beginning of synthesis. During synthesis, primary particles may form aggregates (polycrystals) through chemical reactions or melting. Aggregates can form agglomerates through physical bonds. This is what we want to express, but the drawing of the picture is misleading. We revised the picture and provided the following literature:

“Winkler, J. (2019). Dispersing pigments and fillers. Vincentz Network. ”

+

  1. It will be appropriate to include in Table 1 the size of the particle. Then it is the characterization of the particles by microscopy, but in that table where the basic properties are listed, it is important to include the size.

[Response] Thanks very much for the reviewer’s suggestion. In fact, in different dispersion condition, the particle size is different. Under dry conditions, HFS particles are visible to the naked eye (1-2mm). Under the condition of poor dispersion in water, the particle size is about ~100 μm, while under the condition of good dispersion, the particle size is 130nm. Therefore, it is necessary to specify what conditions are used to disperse the HFS. We attach the HFS sizes observed under the microscope under dry conditions, and the HFS particles observed under good and bad dispersion conditions in water (Figure 6). Also, we have added the particles size obtained in TEM images in the Table 1.

Materials

Type

Surface group

BET-Area /

(m2·g-1

Stacking density /

(g·L-1)

Particle size

HFS

hydrophobic

Dimethylsiloxy

170 ± 30

40

200 ~ 300 nm

  1. Again, regarding redaction, in line 120 authors write: in order to obtain more intuitive particle size results, the microscopic morphology 120 and particle size distribution of HFS were observed by TEM. Intuitive is not the correct word, because when a technique like TEM is used, nothing is intuitive, the image is the reality (under high vacuum conditions), so it could be more accurate to use estimate instead of intuitive. In line 290, the word intuitive is again employed. In line 139, a point has to be replaced by a comma: ´´ dispersant D (anionic polymer dispersant). and dispersant E (non-ionic polymer dispersant´´.

[Response] Thanks for the reviewer’s suggestion. We have revised the statement and the mistakes.

“In order to obtain clearer particle size results, the microscopic morphology and particle size distribution of HFS were observed by TEM. The results are shown in Figure 2.”

“In order to further evaluate the dispersion of HFS in the film, EDS mapping can be used to estimate the characteristic Si elements distribution of HFS in the film section.”

“According to the formula in Table 2, HFS was dispersed by dispersant A (sodium hexametaphosphate), dispersant B (sodium dodecyl benzene sulfonate), dis-persant C (cationic polymer dispersant), dispersant D (sodium polycarboxylate, anionic polymer dispersant), and dispersant E (non-ionic polymer dispersant).”

  1. In line 150 authors state that the former dispersant (polycarboxylate) produces anchorage on the surface of HFS particles through strong polar carboxyl groups. It is not clear how it happens due to the negative surface charge of fumed silica particles.

[Response] Thanks for the reviewer’s question. We have revised our opinion that dispersant (polycarboxylate) produces anchorage on the surface of HFS particles through strong polar carboxyl groups. Actually, HFS has strong adsorption effect on the long chain alkyl of the dispersant (polycarboxylate). The negative charge on the surface of HFS particles prevents HFS particles from approaching and agglomerating with other HFS particles.

“For the former, HFS has strong adsorption effect on the long chain alkyl of the dispersant [35]. The negatively charged carboxyl group of the dispersant is exposed to the solution and forms a solvation chain.”

[35] Venancio, J. C. C.; Nascimento, R. S. V.; Perez-Gramatges, A. Colloidal stability and dynamic adsorption behavior of nanofluids containing alkyl-modified silica nanoparticles and anionic surfactant. J. Mol. Liq. 2020. 308, 113079.

  1. Finally, for SEM characterization, figure 9, it will be more accurate to show images with higher zoom. In this case, the scale bar is 200 nm, while non-agglomerates are in that order under high vacuum conditions, as can be observed from the image. 

[Response] Thanks for the reviewer’s suggestion. After serious consideration, we think that the SEM images shown in Figure 11 are not perfect. However, the agglomerates could be clearly observed in the figure. What’s more, the reason why we designed this experiment is to discuss the dispersion of HFS inside the coating (uniform or uneven).

Reviewer 2 Report

The research paper studies preparation and dispersion performance of hydrophobic fumed silica aqueous dispersion.

It is a very interesting theme. I suggest to authors to make few corrections and additions in the manuscript before its publishing:
1.Last sentence of the abstract contains many mistakes, e.g. "HFS aqueous dispersion can be obtained", abbreviation PDI is not stated....
2. Wrong line numbering
3. First sentence in page 2 - "Chemical" should not be capital.
4. The last paragraph of abstract describes the method of HFS aqueous preparation used in the research. It should not
be stated in the abstract part....but in Materials and Methods
5. The sentence "Among them, the carbon content can indirectly determine the degree of hydrophobic modification of the sample."
   is out of the context.
6. Table 1 - BET-Area is not suitable expression, and surface group should be described in more detail...
7. Figure 1 caption is too long or maybe you forgot to separate the text.
8. Correct the expression: "after stood for 24 h to defoam"
9. Table 3 - "Waterborne polyacrylics latex/ Waterborne polyurethane resin" is unclear - What did you use?
10. 3.1.1 - Results are not shown in Figure 1
11. It is not stated anywhere why specific dispersants (A-E) were chosen for the research.
12. The statement: "The same surface charge leads to electrostatic repulsion" should be explained.
13. Figure 4 - Wrong unit (d.nm)
14. What means "the Z-Average particle size"?
15. Figure 7 - "a", "b" is too big in comparison with the data in graphs.
16. In the abstract you stated that you used "TEM", but in Results you stated "SEM", what type of microscopy did you use?
17. The thickness of coating is not stated.
18. 3.1.4.-room temperature should be specify, also relative humidity.....
19. The conclusion lacks a description of the use of the results in practice.

The manuscript contains quite a lot of grammatical errors

Author Response

Reviewer 2

The research paper studies preparation and dispersion performance of hydrophobic fumed silica aqueous dispersion.

It is a very interesting theme. I suggest to authors to make few corrections and additions in the manuscript before its publishing:

[Response] Thanks very much for the reviewer’s positive comments to our work. The involved issues have been carefully checked and point-to-point responses were given as follows.

  1. Last sentence of the abstract contains many mistakes, e.g. "HFS aqueous dispersion can be obtained", abbreviation PDI is not stated....

[Response] Thanks for the reviewer’s reminder. We have revised the mistakes.

“Under the optimized condition, the HFS aqueous dispersion can be prepared with a particle size of D50=27.2nm. The HFS aqueous dispersion has stable storage stability. Even after storage for 47 d, the particle size still did not change significantly.”

  1. Wrong line numbering

[Response] Thanks for the reviewer’s correction. We have re-checked the articles, and corrected the wrong line numbering.

  1. First sentence in page 2 - "Chemical" should not be capital. 

[Response] Thanks for the reviewer’s reminder. We have revised the mistakes and checked the entire article.

“During the grinding process, the frequent collision of beads effectively breaks the physical bond and chemical bond between nano particles [28].”

  1. The last paragraph of abstract describes the method of HFS aqueous preparation used in the research. It should not be stated in the abstract part....but in Materials and Methods
    [Response] Thanks for the reviewer’s correction. We have revised the abstract and removed the preparation of materials to “2. Materials and Methods”. The new abstract is as follows:

Abstract: Hydrophobic fumed silica (HFS) is a commonly used rheology additive in waterborne coatings. A series experiment was conducted on the HFS dispersing technology in this study. The size and structure of HFS primary particles was observed by transmission electron microscopy (TEM). The measurement results of TEM were D50 = 13.6 nm and D90 = 19.7 nm, respectively. The particle size and dispersion performance of HFS was tested by dynamic light scattering (DLS). Also, the HFS aqueous dispersion was prepared and compounded with waterborne polyacrylic latex and polyurethane resin. The elemental distribution of the coatings was characterized by energy dispersive spectroscopy (EDS). The results showed that the HFS in non-ionic polymer dispersant had the best dispersion performance. The particle size of HFS in the aqueous dispersion is related to the dispersion conditions. Under the optimized condition, the HFS aqueous dispersion can be prepared with a particle size of D50=27.2 nm. The HFS aqueous dispersion has stable storage stability. Even after storage for 47 d, the particle size still did not change significantly.”

  1. The sentence "Among them, the carbon content can indirectly determine the degree of hydrophobic modification of the sample."

[Response] Thanks for the reviewer’s correction. We have deleted this wrong opinion in the article.

  1. Table 1 - BET-Area is not suitable expression, and surface group should be described in more detail...

[Response] Thanks very much for the reviewer’s suggestion. We have fully revised the table, including deleting the BET area and adding the surface group. The revised table is shown as below:

Table 1. Basic properties of HFS

Materials

Type

Surface group

BET-Area /

(m2·g-1

Stacking density /

(g·L-1)

Particle size

HFS

hydrophobic

Dimethylsiloxy

170 ± 30

40

200 ~ 300 nm

  1. Figure 1 caption is too long or maybe you forgot to separate the text.

[Response] Thanks for the reviewer’s correction. We have re-checked the title of Figure 1.

  1. Correct the expression: "after stood for 24 h to defoam" 

[Response] Thanks for the reviewer’s correction. We have revised the sentence.

After left for 24 h to defoam, the coating was poured into a PTFE mold (Depth: ~ 500 μm) and dried at 50 °C for 3 days to obtain the dried films with thickness around 400 ~ 500 μm.”

  1. Table 3 - "Waterborne polyacrylics latex/ Waterborne polyurethane resin" is unclear - What did you use?

 [Response] Thanks for the reviewer’s reminder. Actually, we used both waterborne polyacrylic latex and waterborne polyurethane latex in our study. And we wanted to observe the dispersibility of HFS in different films. We have revised Resin (waterborne polyacrylics latex or waterborne polyurethane resin).

  1. 3.1.1 - Results are not shown in Figure 1

[Response] Thanks for the reviewer’s reminder. In fact, in different dispersion condition, the particle size is different. Under dry conditions, HFS particles are visible to the naked eye (1-2mm). Under the condition of poor dispersion in water, the particle size is about ~100 um, while under the condition of good dispersion, the particle size is 130nm. Therefore, it is necessary to specify what conditions are used to disperse the HFS. After serious consideration, we attach the HFS sizes observed under the microscope under dry conditions, and the HFS particles observed under good and bad dispersion conditions in water (Figure 6). Also, we have added the particles size obtained in TEM images.

Figure 6. Microscope images of HFS in (a) dried powders, (b) 2 wt% dispersant solution, (c) 10 wt% dispersant solution.

Table 1. Basic properties of HFS

Materials

Type

Surface group

BET-Area /

(m2·g-1

Stacking density /

(g·L-1)

Particle size (nm)

HFS

hydrophobic

Dimethylsiloxy

170 ± 30

40

200 ~ 300

  1. It is not stated anywhere why specific dispersants (A-E) were chosen for the research.

 [Response] Thanks very much for the reviewer’s suggestion. In fact, there are many types and structures of dispersants, mainly including non-ionic, cationic and anionic types. The dispersants studied in this paper include commonly used small molecule anionic, polymer anionic, cationic and non-ionic dispersants. It is obviously not possible to discuss all the dispersants, and we will only use a few representative and available dispersants for discussion. In addition, we focus on the influence of the amount of dispersant added and the dispersant process on the slurry preparation, rather than the type of dispersant. We also added this statement in our article:

“In order to study the impact of different dispersants on HFS dispersibility, several representative dispersants were selected.” Page 5, line 153.

  1. The statement: "The same surface charge leads to electrostatic repulsion" should be explained.

[Response] Thanks very much for the reviewer’s suggestion. After serious discussion, we deleted our wrong opinion. We have added this statement and reference in our article:

“For the former, HFS has strong adsorption effect on the long chain alkyl of the dispersant [35]. The negatively charged carboxyl group of the dispersant is exposed to the solution and forms a solvation chain.” Page 5, line 170.

[35] Venancio, J. C. C.; Nascimento, R. S. V.; Perez-Gramatges, A. Colloidal stability and dynamic adsorption behavior of nanofluids containing alkyl-modified silica nanoparticles and anionic surfactant. J. Mol. Liq. 2020. 308, 113079.

  1. Figure 4 - Wrong unit (d.nm)

[Response] Thanks for the reviewer’s correction. We have revised the Figure 5 and Figure 8. The revised figures are shown below:

Figure 5. The relationship between the addition of dispersant E and the particle size

Figure 8. The relationship between the ultrasonic dispersion time and the particle size

  1. What means "the Z-Average particle size"?

[Response] Thanks for the reviewer’s question. Z-average size is the most important and stable data obtained in dynamic light scattering technique. This average particle size is obtained by testing the average scattered light intensity of the sample passing through the solution, and then using a special calculation formula. In our paper, we introduce two results: mean particle size (D50) and Z-Average size. In fact, there is often some difference between the two results. It is common to observe large difference between D50 and Z-Average size:

Salazar, J., Heinzerling, O., Müller, R. H., & Möschwitzer, J. P. Process optimization of a novel production method for nanosuspensions using design of experiments (DoE). International Journal of Pharmaceutics, 2011, 420(2), 395-403.

  1. Figure 7 - "a", "b" is too big in comparison with the data in graphs.

[Response] Thanks for the reviewer’s correction. We have revised the Figure 9. The revised figures are shown below:

Figure 9. Particle size test results under optimal dispersion conditions (a) storage after 0 d, (b) storage after 47 d

  1. In the abstract you stated that you used "TEM", but in Results you stated "SEM", what type of microscopy did you use?

[Response] Thanks for the reviewer’s correction. In fact, we observed the morphology of HFS particles using TEM (Figure 2). Then SEM was used to observe the surface and cross section of the coating (Figure 10, 11, 12). We have revised our statement in article. The revised sentence is shown below:

“The microscopic morphology and particle size of hydrophobic fumed silica were characterized and analyzed through microscope (PH-PG3230, Shanghai, Phenix Op-tics Co., Ltd) and JEOL-2100F trans-mission electron microscope (TEM). The cross-section of the coatings was swept by SEM-EDS energy spectrometer to obtain the element distribution.” Page 4, line 116-120.

  1. The thickness of coating is not stated.

[Response] Thanks very much for the reviewer’s suggestion. In this study, we are only interested in the HFS distribution of the coatings. Therefore, we used a nummular PTFE mold to cast the film without coating on any substrate. We have added some relative information in “2.3. Preparation of waterborne composite film”:

“After left for 24 h to defoam, the coating was poured into a PTFE mold (Depth: ~ 500 μm) and dried at 50 °C for 3 days to obtain the dried films with thickness around 400 ~ 500 μm.” Page 3, line 110-112.

  1. 3.1.4.-room temperature should be specify, also relative humidity.....

[Response] Thanks very much for the reviewer’s suggestion. We have added the below statement in our article:

“In order to further investigate the storage stability of HFS aqueous dispersion after dilution, the dispersive solution was ultrasonic for 5 min, and stayed at constant temperature (25 °C) and humidity (30 %) for 0 d, 3 d and 47 d.” Page 6, line 233-235.

  1. The conclusion lacks a description of the use of the results in practice.

[Response] Thanks very much for the reviewer’s suggestion. We have fully revised the discussion part. We also add more explain about the testing results and what’s the value for the HFS dispersing area.

“In this paper, HFS slurry and aqueous dispersion was prepared, and the dispersion technology was studied. The results of water contact angle showed that the non-ionic dispersant had the best wetting and dispersing effect on HFS particles. To obtain the best dispersion effect, the addition of non-ionic dispersant should be added to 10 wt%. The study on ultrasonic dispersion process indicates that ultrasonic dispersion for 5 min is the most reasonable. Moreover, it is exciting to note that the HFS aqueous dispersion is stable after using a process that combines high speed dispersion and ultrasonic dispersion. After storage for 47 d, the particle size remained at D50=31.0 nm. This shows that a non-abrasive process is feasible for dispersing HFS. This new process may save the energy cost for related enterprises. Moreover, EDS and SEM showed that the dispersion of HFS aqueous dispersion in acrylic resin was uniform and the compatibility was good. This indicates that HFS dispersion has application value.” Page 11, line 355-366.

Reviewer 3 Report

Attached

NA

Author Response

Reviewer 3

The study investigates HFS dispersing technology. Different dispersants were utilized to prepare HFS slurry. Transmission electron microscopy (TEM) and dynamic light scattering (DLS) were utilized to determine particle size and dispersion performance. Results indicated non-ionic polymer dispersant provided the best dispersion performance.

As a reviewer, I find several issues with the manuscript that lead me to recommend rejection.  These issues include the overall clarity of the manuscript, the way the data is presented, and the overall explanation and contribution of the research to the field. To improve the quality of the paper, I recommend the following comments:

[Response] We are glad of having the valuable comments to our work. The involved issues have been carefully checked and point-to-point responses were given as follows. We hope that after fully revision, this paper could satisfy you and would be suitable for publication.

  1. Introduction

Provide a brief overview of the proposed two-step method for preparing HFS slurry. Please explain why this method was chosen and how it addresses the challenges mentioned earlier. It should highlight how this study contributes to the existing knowledge on HFS dispersion and how it advances the field.

[Response] Thanks very much for the reviewer’s suggestion. For clearer descried what this research studying about, we have added more detail about high-speed dispersion and ultrasonic dispersion into our manuscript:

“The energy consumption of ultrasonic dispersion is between grinding and high-speed dispersion [28]. However, the dispersion efficiency of ultrasonic dispersion is much higher than that of high-speed dispersion, which is close to grinding [27]. The combi-nation of high-speed dispersion and ultrasonic dispersion may be able to reach particle size close to milling while reducing energy consumption.” Page 2, line 56-60.

  1. Please provide more background information about hydrophobic fumed silica and its significance in coatings, adhesives, and other composite materials. Elaborate on why its dispersion in water is difficult and how it impacts the application properties.

[Response] Thanks very much for the reviewer’s useful suggestion. To give more clear information about HFS, we have revised the introduction and added some reference into our manuscript:

“Fumed silica is a kind of amorphous particle with a small size, between 10 ~ 40 nm, and extremely large specific surface area [4]. In addition to providing thermal insulation, thickening, thixotropy, and anti-settling [5-8], it also has the effect of reinforcement, color development, and improving powder fluidity [9-13]. Fumed silica has been widely used in coatings, adhesives, and other composite materials [14-19]. Hydrophobic fumed silica (HFS) is aerosol with alkanes, chloromethyl group or other hydrophobic functional groups on surface [20]. Under the joint action of hydrophobic functional groups and hydrogen bonds, a three-dimensional network structure can be formed in a highly polar system, which could provide better rheological(thixotropy) and mechanical properties for waterborne coatings and adhesives compared to other fumed silica [21-23]. But the hydrophobic functional groups also made HFS extremely difficult to disperse in water [24].” Page 1, line 35-41.

Experimental

3.In the materials section, please specify the particle size of the as-purchased HFS, as it appears to be in the range of 200-300 nm according to Figure 2.

[Response] Thanks very much for the reviewer’s suggestion. In fact, in different dispersion condition, the particle size is different. Under dry conditions, HFS particles are visible to the naked eye (1-2mm). Under the condition of poor dispersion in water, the particle size is about ~100 um, while under the condition of good dispersion, the particle size is 130nm. Therefore, it is necessary to specify what conditions are used to disperse the HFS. After serious consideration, we attach the HFS sizes observed under the microscope under dry conditions, and the HFS particles observed under good and bad dispersion conditions in water (Figure 6). Also, we have added the particles size obtained in TEM images.

Table 1. Basic properties of HFS

Materials

Type

Surface group

BET-Area /

(m2·g-1

Stacking density /

(g·L-1)

Particle size (nm)

HFS

hydrophobic

Dimethylsiloxy

170 ± 30

40

200 ~ 300

Figure 6. Microscope images of HFS in (a) dried powders, (b) 2 wt% dispersant solution, (c) 10 wt% dispersant solution.

4.Clarify whether any type of defoamer was used during slurry preparation. If not, elaborate on how foam formation was controlled throughout the process.

[Response] Thanks for the reviewer’s question. The foam was not observed after the slurry was left for 2h after preparation. We have added the below statement in our article:

“During the dispersion process, the slurry was kept at a constant temperature (25 °C) via a water-cooling equipment. The HFS slurries were stored for 2 h for defoaming.” Page 3, line 99-101.

5.Provide information about the temperature during the shearing process.

[Response] Thanks for the reviewer’s question. In experiment, we carried out strict temperature control using cooling equipment. We have added the below statement in our article:

“During the dispersion process, the slurry was kept at a constant temperature (25 °C) via a water-cooling equipment. The HFS slurries were stored for 2 h for defoaming.” Page 3, line 99-101.

6.Improve Fig. 1 schematic by including additional details such as the shear speed of HFS slurry dispersion, pH conditions, and the duration of ultrasonication process. For the coating preparation, mention other relevant details.

Specify the substrate used for the coating and provide a clear procedure for coating the HFS slurry. If it was a laminate, make sure to mention it explicitly.

[Response] Thanks very much for the reviewer’s useful suggestion. We have added more information in Figure 1, such as pH value, rotation speed and so on. We also noticed that we do not mention how we prepared the coating films in our manuscript.

In fact, in this study we are only interested in the HFS distribution of the coating cross-section. Therefore, we used a PTFE mold to cast the films without coating on any substrate.

Figure 1. The schematic illustration of HFS aqueous dispersion and coatings preparation

“The rotation speed was set as 800 rpm, and stirred the coatings for 10 min. After left for 24 h to defoam, the coating was poured into a PTFE mold (Depth: ~ 500 μm) and dried at 50 °C for 3 days to obtain the dried films with thickness around 400 ~ 500 μm.” Page 3, line 109-112.

  1. Include the types of waterborne resins utilized in the study within the experimental section.

[Response] Thanks for the reviewer’s suggestion. Actually, we have included the resin types in Tabel 3.

Table 3. Formula of HFS composite coating

Materials

Mass fraction / %

Resin (waterborne polyacrylics latex or waterborne polyurethane resin)

98.0

HFS aqueous dispersion (2 % HFS)

2.0

Results and Discussion

  1. Present the experimental results in a more comprehensive and interpretable manner. Include context and analysis to elucidate the findings' implications for the research field.

[Response] Thanks very much for the reviewer’s suggestion. We have fully re-checked the article and revised our manuscript according to the below suggestion. We hope this new article will satisfy you.

9.Revise the caption for Figure 2 to be more specific, indicating all component figures (a, b, c). Clearly specify whether the image represents as-procured HFS powder or HFS slurry to avoid ambiguity.

[Response] Thanks very much for the reviewer’s suggestion. We have revised the title of Figure 2.

Figure 2. TEM images in different magnification (a) 30,000×, (b) 60,000×, (c) particle distribution of HFS dried powder and (d) the schematic illustration of HFS agglomerates formation.

10.Please include references that explain the chemical bonds responsible for the size range of 20-100 nm and the physical bonds leading to the increase in size to 200-300 nm. Assign image number to the schematic showing physical/chemical bonds. Ensure that the direction of the arrow in the image accurately reflects the reduction of agglomerates from 100-200 nm to a final size of 10-20 nm. Clearly state in the manuscript that the agglomerate size remained unchanged at 100-200 nm even after dispersion. Provide more details about the individual component images in the description of Figure 2, rather than only focusing on Figure 2c.

[Response] Thanks very much for the reviewer’s suggestion. We have collected some reference that could confirm the size of primary particles, aggregates (> 100 nm) and agglomerates (> 200 nm). The references are shown as below:

[32] Du, A.; Zhou, B.; Zhang, Z.; Shen, J. A special material or a new state of matter: a review and reconsideration of the aerogel. Materials. 2013, 6(3), 941–968.

[33] Linhares, T.; de Amorim, M. T. P.; Durães, L. Silica aerogel composites with embedded fibres: a review on their preparation, properties and applications. J. Mat. Chem A 2019. 7(40), 22768–22802.

[34] Mazrouei-Sebdani, Z.; Naeimirad, M.; Peterek, S.; Begum, H.; Galmarini, S.; Pursche, F.; Baskin E.; Zhao, S.; Gries, T.; Malfait, W. J. Multiple assembly strategies for silica aerogel-fiber combinations–A review. Mater. Design 2022. 111228.

We also added more clearer statement about particle size:

“The TEM image in Figure 2 can more clearly characterize the shape and particle size of HFS. It can be seen from Figures 2(a) and (b) that HFS particles have many pore structures. HFS agglomerates have particle size around 200 to 300 nm [32]. It is formed by serval 50 ~ 150 nm aggregates which connected by "bridges" [33-34]. These so-called “bridges” are actually formed by physical bonds between the edges of the aggregate and other aggregates (Figure 2 (d)).” Page 4, line 135-140.

11.On page 5, line 137, the statement "According to the formula in Table 2," the dispersant composition is indicated to vary from 4-10%. However, in Figure 3, it shows that the dispersant composition was 6% for that specific case. Please revise the sentence for clarity.

[Response] Thanks very much for the reviewer’s suggestion. Actually, when the addition of dispersant was lower than 6 wt%, no matter which dispersant was used, the HFS slurry will seem like gel. Only when the addition of dispersant was equal to 6 wt%, the dispersibility of HFS in dispersant D or E solution was much better than other dispersant solution. Therefore, 6 wt% addition of dispersant was selected to finish the experiment in 3.1.2. We have added the below statement in our article:

“When the addition of dispersant was lower than 6 wt%, no matter which dispersant was used, the HFS slurry will seem like gel.” Page 5, line 156-157.

12.Caption of Figure 3:  Provide information about the dispersant. Classify all dispersants (A-E) as cationic or ionic, or non-ionic.

[Response] Thanks for the reviewer’s correction. We have revised the title of Figure 3.

Figure 3. HFS slurry prepared with A (anionic dispersant); B (anionic dispersant); C (cationic polymer dispersant); D (anionic polymer dispersant); E (non-ionic polymer dispersant).

13.Support the statement "However, the former dispersant produces anchorage on the surface of HFS particles through strong polar group carboxyl. The same surface charge leads to electrostatic repulsion" with suitable references.

[Response] Thanks for the reviewer’s correction. We have revised our opinion that dispersant (polycarboxylate) produces anchorage on the surface of HFS particles through strong polar carboxyl groups. Actually, HFS has strong adsorption effect on the long chain alkyl of the dispersant (polycarboxylate). The negative charge on the surface of HFS particles prevents HFS particles from approaching and agglomerating with other HFS particles.

“For the former, HFS has strong adsorption effect on the long chain alkyl of the dispersant [35]. The negatively charged carboxyl group of the dispersant is exposed to the solution and forms a solvation chain.”

[35] Venancio, J. C. C.; Nascimento, R. S. V.; Perez-Gramatges, A. Colloidal stability and dynamic adsorption behavior of nanofluids containing alkyl-modified silica nanoparticles and anionic surfactant. J. Mol. Liq. 2020. 308, 113079.

14.Page 5, line 142-163, the paragraph should be rephrased. Ensure a clear and coherent explanation of why dispersant E demonstrated superior properties over D.

[Response] Thanks very much for the reviewer’s useful suggestion. After serious consideration, we think it is necessary to add more data to explain why dispersant E has the best dispersibility among dispersant A, B, C, D and E. Therefore, we designed an experiment in “3.1.2. The HFS dispersibility with different dispersants” as you recommend:

Figure 4. The water contact angle of different dispersant solution (6 wt%) on the surface coated with HFS powder. A (anionic dispersant); B (anionic dispersant); C (cationic polymer dispersant); D (anionic polymer dispersant); E (non-ionic polymer dispersant).

“In order to accurately distinguish the wetting state of different dispersant solutions on the surface of HFS particles, the water contact angle was measured. Due to the alkylation of the HFS surface, water is totally unable to wet the HFS surface, and the water contact angle is even larger than 170 °. After the addition of dispersant (A, B, C), the wetting condition was significantly improved, and the contact angle was reduced to 110 ~ 130 °. With the addition of dispersant D, the contact angle was further reduced to 56 °, indicating a transition from non-wettable to wettable. The water contact angle of non-ionic dispersant E is only 46 °, and the wetting performance is significantly better than that of dispersant D. What’s more, large number of anionic surfactants are easy to aggregate in the waterborne system and form ion channels inside the coatings, which will affect the initial water resistance and long-term durability of the coatings, especially for waterborne industrial coatings with high initial water resistance requirements [39]. Therefore, it is reasonable to use non-ionic dispersant (E) to disperse HFS.” Page 6, line 176-188.

This testing method could be referred in the follow reference:

[31] Tai, X.; Ma, J.; Du, Z.; Wang, W. The facile preparation for temperature sensitive silica/PNIPAAm composite microspheres. Appl. Surf. Sci. 2013, 268, 489–495.

15.In the caption and text for Figure 4, specify which dispersant (A, B, C, D, or E) is being used for better context and understanding.

[Response] Thanks very much for the reviewer’s useful suggestion. We have revised the title of Figure 5.

Figure 5. The relationship between the addition of dispersant E and the particle size

16.Consider qualitative methods for assessing dispersibility, such as calculating the surface energy of the dispersion medium using a contact angle goniometer.

[Response] Thanks very much for the reviewer’s useful advice. We have added an experiment for explain why dispersant E has the best dispersibility among dispersant A, B, C, D and E. However, it seems that measuring the water contact angle of the dispersant solution on the HFS particles could explains this directly.

17.The statement "In contrast, HFS was completely unable to disperse without the use of dispersant" lacks support from Figure 4, and its interpretation is confusing. Please consider revising this statement to accurately reflect the findings.

[Response] Thanks very much for the reviewer’s useful advice. In fact, as shown in our complementary experiment, the water contact angle carried on the surface of HFS is larger than 170°. Water could not wet the surface of HFS at all. Therefore, it is impossible to test the particle size of HFS via DLS.

18.Page 6, line 173-188: provide relevant references to substantiate the points.

[Response] Thanks very much for the reviewer’s useful suggestion. To support our argument, we have added the following literature:

[41] Khalkhal, F.; Negi, A. S.; Harrison, J.; Stokes, C. D.; Morgan, D. L.; Osuji, C. O. Evaluating the dispersant stabilization of colloidal suspensions from the scaling behavior of gel rheology and adsorption measurements. Langmuir, 2018. 34(3), 1092-1099.

[42] Hu, S.; Li, J.; Liu, K.; Chen, Y. Comparative study on distribution characteristics of anionic dispersants in coal water slurry. Colloids Surfaces A 2022. 648, 129176.

19.Considering the optimized dispersant wt% in Figure 4 is 10%, it is essential to investigate whether further increases in dispersant percentage (>10%) can lead to a reduction in particle size. Additionally, the particle size of 130 nm after adding 10% dispersant exceeds the physical bond range (100 nm), requiring further discussion in the manuscript.

[Response] Thanks very much for the reviewer’s suggestion. According to this suggestion, we have added more data in Figure 5. Regarding to the second question, it is common to observe the agglomerates which particle size larger than 100 nm (especially for those fillers with low surface energy or with large particle size). We have added the reason why the HFS particle size in aqueous dispersion (adding 10 wt% dispersant) is still larger than 100 nm in “3.1.3. The impact of dispersant dosage on the HFS dispersibility”:

“(1) When the addition of dispersant increased, the adsorption of dispersant molecules on the HFS particles surface also increased. At the same time, the physical bonds strength between aggregates have been weakened. As a result, HFS become more and more easily dispersed, and the particle size becomes smaller after ultrasonic dispersion. (2) The dispersant reached saturation adsorption on the HFS surface [39-40]. The ex-cess dispersant will form micelles in the solution and have no effect on particle disper-sion. Therefore, the particle size no longer decreases. (3) The strong chemical bonds between primary particles could not be broken by adding dispersant. But the disper-sant successfully broke the physical bonds. This study also mean that the chemical bonds cannot be broken by high-speed dispersion and ultrasonic dispersion.” Page 6, line 203-213.

20.Clarify in the manuscript in which media the HFS slurry containing 10 wt% dispersant E was diluted.

[Response] Thanks for the reviewer’s question, the slurry is diluted in water. The relative information is shown in “2.2. Preparation of HFS slurry and aqueous dispersion”. However, to make clear statement, we also add the relative information in Figure 1.

Figure 1. The schematic illustration of HFS aqueous dispersion and coatings preparation

21.Emphasize the interesting finding of the coating's stability up to 47 days without any stabilizing agent in the manuscript to highlight its significance.

[Response] Thanks very much for the reviewer’s suggestion. To enhance the important finding of 47 d stability, we added some statement in the abstract and conclusion of our manuscript:

“Under the optimized condition, the HFS aqueous dispersion can be prepared with a particle size of D50=27.2nm. The HFS aqueous dispersion has stable storage stability. Even after storage for 47 d, the particle size still did not change significantly.” Page 1, line 19-21.

“Moreover, it is exciting to note that the HFS aqueous dispersion is stable after using a process that combines high speed dispersion and ultrasonic dispersion. After storage for 47 d, the particle size remained at D50=31.0 nm. This shows that a non-abrasive process is feasible for dispersing HFS.” Page 11, line 358-361.

22.What is the difference in particle size plotted in Figure 4 and Figure 6? Figure 6 seems to be before and after high-speed dispersion process. When was the measurements for particles were taken for Figure 4?

[Response] Thanks very much for the reviewer’s question. Actually, both Figure 5 and Figure 8 were obtained via DLS after the HFS was dispersed into aqueous dispersion, but different ultrasonic dispersion time and dispersant addition. We have added the condition that we used to tested Z-Average size:

“In order to study how dispersant E addition affect the particle size of HFS aqueous dispersion, the dilution ratio was set at 10 times. Five minutes ultrasonic dispersion time was performed to obtain sufficient dispersion energy.” Page 6, line 190-192.

“To confirm the impact of ultrasonic dispersing time on the particle size of HFS aqueous dispersion, a study was conducted on this. Dispersant usage and dilution ratio were set at 10 wt% and 10x, respectively.” Page 8, line 247-249.

23.Page 6, line 230-239: authors mentioned that after ultrasonic dispersion, the particles may agglomerate again due to energy loss in the system. To address this concern, I suggest the authors highlight the specific procedures they have implemented to prevent particle agglomeration for an extended duration. I recommend adding this information to the manuscript for clarity.

[Response] Thanks very much for the reviewer’s suggestion. From the storage test, it can be seen that the Z -Average size only increased from 126 nm to 129 nm after 47 d. In the particle size study in this paper, we performed the particle size test immediately after completing the ultrasonic dispersion. So, we think that short-term (1-24h) storage will not affect the dispersion of HFS in aqueous dispersion. We have added the below statement in our manuscript:

“The particle size test was performed on the HFS aqueous dispersion just after completing the ultrasonic dispersion.” Page 4, line 123-124.

24.Again, it is difficult to understand, at which stage of experiment, data for Figure 4 and Figure 6 taken. I assumed that in Figure 4 against 10% dispersant, the particle size was optimized ~130 nm. But, Figure 7a represents, the optimized particle size is in 27.2 nm. To avoid confusion, please clarify the stages at which the particle size data were obtained for each figure before discussing their results in the manuscript.

[Response] Thanks very much for the reviewer’s question. The particle size shown in Figure 9 was obtained under the optimized process parameter (dilution 10×, ultrasonic dispersion time 10 min), different from Figure 5 and Figure 8. As you mention in the question, they have “different particle size”. Under optimized process parameter, the particle size shown in Figure 5 is 130 nm (Z-Average size), another shown in Figure 9 is 27.2 nm (D50 or mean particle size). Actually, both Z-Average size and D50 were obtained under the same time, but calculated by different equation of the analysis software. It is very common to observe some difference between Z-Average size and D50.We consider that many other researchers may care about the D50, particle size distribution and Z-Average size. Even some researchers seem them as same important results. Therefore, we show all the results in Figure 9. We also add the below statement and add the Z-Average size results in Figure 9:

“The optimized process parameters (dilution 10x, ultrasonic dispersion time 10 min) obtained from the above analysis were used to prepare HFS aqueous dispersion.” Page 9, line 286-287.

Figure 9. Particle size test results under optimal dispersion conditions (a) storage after 0 d, (b) storage after 47 d

It is common to observe large difference between D50 and Z-Average size. Here we show a reference:

Salazar, J., Heinzerling, O., Müller, R. H., & Möschwitzer, J. P. Process optimization of a novel production method for nanosuspensions using design of experiments (DoE). International Journal of Pharmaceutics, 2011, 420(2), 395-403.

25.The discussion section does not provide a comprehensive explanation of the findings presented in the paper. Please include a more detailed analysis and interpretation of the results, providing insights into their significance and implications for the research field.

[Response] Thanks very much for the reviewer’s suggestion. We have fully revised the discussion part. We also add more explain about the testing results and what’s the value for the HFS dispersing area.

“In this paper, HFS slurry and aqueous dispersion was prepared, and the dispersion technology was studied. The results of water contact angle showed that the non-ionic dispersant had the best wetting and dispersing effect on HFS particles. To obtain the best dispersion effect, the addition of non-ionic dispersant should be added to 10 wt%. The study on ultrasonic dispersion process indicates that ultrasonic dispersion for 5 min is the most reasonable. Moreover, it is exciting to note that the HFS aqueous dispersion is stable after using a process that combines high speed dispersion and ultrasonic dispersion. After storage for 47 d, the particle size remained at D50=31.0 nm. This shows that a non-abrasive process is feasible for dispersing HFS. This new process may save the energy cost for related enterprises. Moreover, EDS and SEM showed that the dispersion of HFS aqueous dispersion in acrylic resin was uniform and the compatibility was good. This indicates that HFS dispersion has application value.” Page 11, line 355-366.

26.The manuscript lacks information about the properties of the coating, its performance, applicability, and the process involved in its development. To improve the clarity of the paper, I suggest including details about the coating's characteristics, its performance in relevant applications, and the purpose behind its preparation.

[Response] Thanks very much for the reviewer’s suggestion. In fact, the purpose of providing cross-section SEM images of films is to discuss whether HFS aqueous dispersion will be also evenly dispersed in coatings. In fact, the experimental results were as expected.

If necessary, we can delete “3.2. Dispersibility of HFS in coatings”, and add more results about high-speed dispersing. So that the readers could pay more attention to the dispersion technology of HFS.

27.Figure 7: Full form of PDI

[Response] Thanks very much for the reviewer’s suggestion. PDI means polydispersity index, which is relative to the particle size distribution. We have added the full form of PDI in article:

“The particle size and polydispersity index (PDI) of the nanoparticles in the dispersion were tested by the Zetasizer Nano-S90 laser particle size analyzer of Malvern Company.” Page 4, line 125.

Round 2

Reviewer 2 Report

Dear authors,

you incorporated all of my comments in the paper. I think, it can be published.

Only minimum editing of English language is required.

Reviewer 3 Report

The manuscript can be accepted. All comments are addressed. One mibor suggestion. "Discussion" section can be renamed as "Conclusion."

NA